# Preparation and Application of Electrochemical Horseradish Peroxidase Sensor Based on a Black Phosphorene and Single-Walled Carbon Nanotubes Nanocomposite

**DOI:** 10.3390/molecules27228064

**Published:** 2022-11-20

**Authors:** Xiaoqing Li, Lisi Wang, Baoli Wang, Siyue Zhang, Meng Jiang, Wanting Fu, Wei Sun

**Affiliations:** 1Key Laboratory of Functional Materials and Photoelectrochemistry of Haikou, Key Laboratory of Water Pollution Treatment and Resource Rouse of Hainan Province, College of Chemistry and Chemical Engineering, Hainan Normal University, Haikou 571158, China; 2College of Health Sciences, Shandong University of Traditional Chinese Medicine, Jinan 250355, China; 3College of Health Sciences, Hainan Technology and Business College, Haikou 570102, China

**Keywords:** electrochemical biosensor, horseradish peroxidase, black phosphorene, single-walled carbon nanotubes, trichloroacetic acid, nitrite, hydrogen peroxide

## Abstract

To design a new electrochemical horseradish peroxidase (HRP) biosensor with excellent analytical performance, black phosphorene (BP) nanosheets and single-walled carbon nanotubes (SWCNTs) nanocomposites were used as the modifier, with a carbon ionic liquid electrode (CILE) as the substrate electrode. The SWCNTs-BP nanocomposite was synthesized by a simple in situ mixing procedure and modified on the CILE surface by the direct casting method. Then HRP was immobilized on the modified electrode with Nafion film. The electrocatalysis of this electrochemical HRP biosensor to various targets was further explored. Experimental results indicated that the direct electrochemistry of HRP was realized with a pair of symmetric and quasi-reversible redox peaks appeared, which was due to the presence of SWCNTs-BP on the surface of CILE, exhibiting synergistic effects with high electrical conductivity and good biocompatibility. Excellent electrocatalytic activity to trichloroacetic acid (TCA), sodium nitrite (NaNO_2_), and hydrogen peroxide (H_2_O_2_) were realized, with a wide linear range and a low detection limit. Different real samples, such as a medical facial peel solution, the soak water of pickled vegetables, and a 3% H_2_O_2_ disinfectant, were further analyzed, with satisfactory results, further proving the potential practical applications for the electrochemical biosensor.

## 1. Introduction

The direct electrochemistry of redox enzymes or redox proteins can be used for the investigation of the electron transfer mechanism in the biological system, which broadens its applications in areas such as disease diagnosis, biosensing, etc. [1,2]. As a type of glycoprotein with iron porphyrin as the electroactive center [3], horseradish peroxidase (HRP) has been widely used in electrochemical biosensors. However, the electroactive centers of HRP are deeply embedded inside the biostructure of the protein, making it difficult to realize the direct electron transfer with the substrate electrode. Therefore, great efforts have been made to exploit a high performance interface to improve the electron transfer rate of HRP. Niu et al., used a platinum-3D graphene oxide aerogel nanocomposite as a modifier for the direct electrochemistry and electrocatalysis of HRP [4]. Liu et al., applied porous Co_3_O_4_ nanosheets and a reduced graphene oxide composite modified electrode for the immobilization of HRP, which was further used for nitrite sensing [5]. Horst et al., designed an HRP biosensor based on a bismuth-silver bimetallic nanocomposite for the detection of H_2_O_2_ with a wide linearity range and a low detection limit [6]. With the development of nanoscience, various nanomaterials have been designed to construct a new electrochemical HRP sensor, with excellent performance.

In recent years, two-dimensional (2D) nanostructure materials have attracted significate research attention owing to their unique electronic, optical, and mechanical properties, which can be used in different fields [7,8,9,10]. As a new 2D nanomaterial, black phosphorene (BP) exhibits a layered structure, and each layer of P atoms is connected by strong intralayer P-P bonding and weak interlayer van der Waals forces [11,12]. Due to its large specific surface area, small electric resistance, high catalytic ability, etc., BP shows a high potential applicability in the field of electrochemical biosensing [13]. Despite its advantageous properties, studies on BP in the field of electrochemical sensing are still lacking. Moreover, BP is easily degraded with water and oxygen in ambient environments [14,15], which causes BP based sensors to be unstable and hinders their applicability. Therefore, great efforts have been made to improve the practical application of BP. Different methods, included packaging, functionalization, liquid surface passivation, and doping, have been used to improve the stability of BP with different usages [16,17,18]. Among these methods, BP coupled with other materials to form composites, is a method often used to increase chemical stability and extend usage fields. For example, Zhao et al., used polylysine-modified BP nanosheets to immobilize hemoglobin (Hb) on a glassy carbon electrode (GCE) for the detection of H_2_O_2_ [19]. Zhuge et al., used BP, ionic liquid, and a poly (diallyldimethylammonium chloride) composite-modified GCE for nitrite analysis [20]. Shi et al., synthesized nitrogen-doped carbonized polymer dots anchoring few-layer BP, and constructed an electrochemical DNA sensor for the determination of Escherichia coli O157: H7, which exhibited excellent electrochemical performance and good stability [21]. The interaction of BP and other components can be originated from the covalent action, π–π stacking interaction, or the hydrogen bonding that exists between the two, which led to the synergistic effects, with increased stability.

As a commonly used carbon nanomaterial, carbon nanotubes (CNTs) exhibit many excellent advantages, such as high electrical conductivity, large specific surface area, rapid electrode kinetics, excellent chemical stability, and good electrocatalytic properties [22]. Single-walled carbon nanotubes (SWCNTs) can be considered as one layer of graphene [23], and they are often used to construct electrochemical biosensors with improved current responses. Sun et al., fabricated an Hb and SWCNTs modified electrode and investigated the electrocatalytic reduction of trichloroacetic acid (TCA) [24]. Deshmukh et al., prepared an ethylenediaminetetraacetic acid chelating ligand-modified polyaniline (PANI) and SWCNTs-based nanocomposite (PANI/SWNCTs) to detect copper (II), lead (II), and mercury (II) ions, with a low detection limit [25]. Zhao et al., designated multi-layers of SWCNTs and protein film, with good electrocatalytic properties toward the reduction of oxygen and hydrogen peroxide [26].

In this study, we developed a highly sensitive electrochemical sensor based on an SWCNTs-BP nanocomposite formed by the electrostatic interaction between the amino groups of SWCNTs and the negative charge of BP. The presence of SWCNTs could suppress the surface oxidation of BP and improve the stability and dispersibility of BP during aqueous-phase exfoliation. By leveraging the biocompatibility of BP and the high catalytic activity and large surface area of SWCNTs, the nanocomposite was modified on the surface of a carbon ionic liquid electrode (CILE) with the subsequent horseradish peroxidase (HRP) immobilization by Nafion (Figure 1). CILE is a kind of carbon paste electrode that uses ionic liquid as the modifier and binder. Due to the best electrochemical properties of ionic liquid, such as higher ionic conductivity and good electrochemical stability, CILE has been proved to exhibit advantages including high conductivity, wide electrochemical windows, anti-fouling capability, and inherent electrocatalytic capacity [27,28,29,30,31]. Therefore, CILE is selected as the substrate electrode for further modification in this paper. The prepared sensor (Nafion/HRP/SWCNTs-BP/CILE) exhibited synergistic effects to improve the electrochemical performance and electrocatalytic ability to TCA, NaNO_2_, and H_2_O_2_. Further, this enzymatic sensor showed satisfactory results for real samples analysis, which demonstrated its potential applicability in electrochemical biosensing.

## 2. Results and Discussion

### 2.1. Characterization of SWCNTs-BP Nanocomposite

As shown in Figure 1, the scanning electron microscopy (SEM) image demonstrated that the dispersion of black phosphorus nanoplates (BPNPs) presented a multi-layered flake structure, and the layers were stacked together (Figure 1A). The SWCNTs were highly tangled with each other, showing typical nanotube features (Figure 1B). A transmission electron microscopy (TEM) image of BP after exfoliation is shown in Figure 1C, which confirmed the ultra-thin nanosheet structure of BP. From the HRTEM image of BP (inset of Figure 1C), the lattice spacing of 0.294 nm can be clearly observed, confirming that the crystal lattice of BP was not distorted during the exfoliation process [32]. Figure 1D,E display the SEM of SWCNTs-BP at different amplifications. BP exhibited a few-layer flake appearance, where the SWCNTs were cross-linked and intertwined with each other and adhered on the surface of BP. The EDS images of the SWCNTs-BP nanocomposite (Figure 1F–H) proved the existence of the elements of C, O, and P, indicating that BP and SWCNTs were cross-linked and fused together.

X-ray photoelectron spectroscopy (XPS) was used to analysis the chemical states and surface compositions of the SWCNTs-BP nanocomposite. The XPS survey spectrum clearly indicated the existence of elemental P and C in SWCNT-BP (Figure 2A). Figure 2B shows the XPS spectrum of P 2p of the BP, in which three peaks located at 130.0 eV, 130.8 eV, and 134.1 eV were ascribed to P 2p3/2, P 2p1/2, and phosphorus oxide (PxOy), respectively [33,34]. However, in the P 2p XPS spectrum of the SWCNTs-BP nanocomposite (Figure 2C), all the binding energy of P 2p3/2, P 2p1/2, and phosphorus oxide (PxOy) shifted to the high energy compared to those of the BP, showing that the SWCNTs on the surface of BP occupied the active site of the P element and reduced the binding ability of P and O [35]. Therefore, the presence of SWCNTs could improve the stability of BP.

Raman spectroscopy was used to verify the successful exfoliation of BP and characterize the phosphorene layer (Figure 2D). The Raman spectrum was also evaluated for BP and SWCNTs to facilitate a better comparison. As for BP, three typical peaks at 361.2 cm^−1^, 439.9 cm^−1^, and 467.1 cm^−1^ appeared, which were attributed to the $A_{g}^{1}$, B_2g_ and $A_{g}^{1}$ modes of BP, respectively [36]. For SWCNTs, the typical peak appeared at 471.8 cm^−1^, which was ascribed to $A_{g}^{1}$, whereas for the SWCNTs-BP nanocomposite, similar signals could be found, indicating that BP maintained the physical characteristics of BPNPs. However, the three typical peaks slightly blue-shifted at about 0.6 cm^−1^, 0.8 cm^−1^ and 1.2 cm^−1^, respectively, which confirmed the reduced number of layers [37,38]. Moreover, a new peak appeared at 272.5 cm^−1^, indicating the interaction between BP and the SWCNTs [39].

### 2.2. Electrochemical Properties of the Modified Electrodes

Cyclic voltammograms of different modified electrodes in a 1.0 mmol/L K_3_[Fe(CN)_6_] and 0.5 mol/L KCl mixture solution were recorded, with the results shown in Appendix A. A pair of symmetric redox peaks from Fe(III)/Fe(II) were observed on all the electrodes, and the redox peak currents increased gradually with the presence of SWCNTs (curve b), BP (curve c), and SWCNTs-BP (curve d) on the CILE surface. The electrochemical parameters of different modified electrodes are listed in Table 1, with the peak-to-peak separation (ΔEp) and the ratio of the cathodic (Ipc) to anodic (Ipa) current calculated. The presence of SWCNTs-BP on the electrode surface resulted in the largest response, which could be attributed to the synergetic enhancement effect of two components, including the good conductivity, high specific surface area, and layered structure.

The effect of the scan rate on the electrochemical responses was further investigated, with the curves shown in the inset of Appendix A. In the range from 0.05 to 1.0 V/s, the redox peak currents showed a good linear relationship with υ^1/2^ (Appendix A). The effective surface area (A) of Nafion/SWCNTs-BP/CILE was calculated as 0.309 cm^2^ by the Randles–Sevcik equation [40], indicating the coexistence of SWCNTs and BP on the surface of CILE, with a specific structure that could increase the effective electrode area.

### 2.3. Direct Electrochemical Behavior of HRP

Figure 3 shows electrochemical characterizations of different modified electrodes in a 0.1 mmol/L PBS using cyclic voltammetry (CV). On bare CILE (curve a), stable background curves could be observed in the scan potential range. On Nafion/HRP/CILE (curve b), a pair of redox peaks appeared, ascribed to the direct electron transfer of the HRP Fe(III)/Fe(II) active center. When SWCNTs (curve c) or BP (curve d) were modified on the electrode, all the redox peak currents were greatly enhanced, which was ascribed to the presence of high conductive SWCNTs or BP on the electrode surface that offered a fast electron conducting pathway for HRP to accelerate electron transfer. Furthermore, on the SWCNTs-BP and HRP-modified electrode (curve e), the electrochemical response was larger than all the others, indicating that the SWCNTs-BP nanocomposite, with a large specific surface area, good biocompatibility and excellent conductivity, was beneficial for the electron transfer between HRP Fe(III)/Fe(II) and the electrode. HRP is a redox enzyme commonly used in electrochemical enzyme biosensors, since it contains iron porphyrin as the electroactive center, composed of the HRP Fe(III)/Fe(II) redox couple. However, direct electron transfer of Fe(III)/Fe(II) is difficult to realize due to the deep burial of the active center in the protein structure and its unfavorable orientation. Therefore, the presence of a higher conductive SWCNTs-BP nanocomposite on the electrode surface is beneficial for accelerating the electron transfer of the Fe(III)/Fe(II) redox couple.

The buffer pH can affect the cyclic voltammetric behaviors of HRP, and Figure 3B shows the cyclic voltammograms of Nafion/HRP/SWCNTs-BP/CILE at different pH levels from 2.0 to 9.0. A pair of symmetrical redox peaks could be observed under different pH values, and the redox peak potentials were negatively shifted with the increase in buffer pH, indicating that protons were involved in the electrochemical process. A good linear regression relationship was achieved between the formal potential (E^0′^) and pH with the equation: E^0′^(V) = −0.0443 pH − 0.0881 (γ = 0.995) (Figure 3C, curve a). The slope value of −44.3 mV/pH was a little smaller than the theoretical value of 59.0 mV/pH [41], reflecting the same proton and electron involved in the transfer process. The plot of the oxidation peak current against the pH value is shown in Figure 3C (curve b), which increased from 2.0 to 4.0, and then decreased from 4.0 to 9.0, with the maximum redox peak current at pH 4.0, which was selected as the supporting electrolyte for determination.

The influence of the scan rate on the electrochemical responses of Nafion/HRP/SWCNTs-BP/CILE in pH 4.0 PBS was further investigated from 0.05 to 1.0 V/s. The nearly symmetrical redox peak currents appeared at different scan rates, and the peak-to-peak separation increased with the scan rate (Figure 3D), proving a quasi-reversible electrode process. Two straight lines were calculated with the equations: Ipc (mA) = 0.369 υ (V/s) + 0.0278 (γ = 0.994) and Ipa (mA) = −0.327 υ (V/s) − 0.0278 (γ = 0.995), manifesting a typical surface-controlled electrochemical process. Meanwhile, the relationships of redox peak potentials (Ep) with lnυ were confirmed with two straight lines as Epc (V) = −0.0433 lnυ (V/s) − 0.273 (γ = 0.997) and Epa (V) = 0.0391 lnυ (mV/s) − 0.144 (γ = 0.993), respectively. According to Laviron’s equation [42,43], the electrochemical parameters were calculated with the electron transfer coefficient (α) and electron transfer number (*n*) as 0.474 and 1.14, respectively. The heterogeneous electron transfer rate constant (*k_s_*) was calculated to be 6.34 s^−1^, which was larger than some reported values of 3.35 s^−1^ [4], 1.01 s^−1^ [44], and 0.94 s^−1^ [45], indicating a fast electron transfer reaction.

### 2.4. Electrocatalytic Activity

As a typical redox enzyme, HRP exhibits excellent catalytic activity, and it has been widely used in enzyme sensors for various analytical targets. TCA is extensively used in herbicides and preservatives, and is potentially deleterious to environmental pollutants owing to its extensive use in industry, biochemistry, agriculture, and public health fields [46,47]. Nitrite is widely used in food products as a preservative, which can become poisonous to human beings at high concentrations, and likely results in cancer [48,49]. Hydrogen peroxide (H_2_O_2_) is a by-product of reactive oxygen metabolism and is a key regulator of a variety of oxidative stress-related states, which are related to the development of various diseases, including cancer [50,51]. Therefore, peroxidase-based electrochemical nano-biosensors have been established for the sensitive analysis of TCA, NaNO_2_, and H_2_O_2_.

The electrocatalytic activity of Nafion/HRP/SWCNTs-BP/CILE towards TCA was investigated in pH 4.0 PBS, with the results shown in Figure 4A. The catalytic reduction peak current increased gradually with the addition of TCA, and the corresponding oxidation peak current in the cyclic voltammogram decreased and eventually disappeared, which was the characteristic electrocatalysis of TCA, with the following catalytic mechanism [44,52].
HRP Fe(III) + H^+^ + e^−^ ↔ HRP Fe(II)
2[HRP Fe(II)] + Cl_3_CCOOH + H^+^ → 2[HRP Fe(III)] + Cl_2_CHCOOH + Cl^−^

The TCA concentration and the reduction peak current had a good linear relationship from 1.0 to 190.0 mmol/L, with the equation of Iss (mA) = 0.0083 C (mmol/L) + 0.0201 (γ = 0.992), and from 190.0 to 810.0 mmol/L, with the equation of Iss (mA) = 0.0025 C (mmol/L) + 1.096 (γ = 0.995) (Figure 4B). The detection limit was obtained as 0.33 mmol/L (3σ).

The electrocatalytic reduction of NaNO_2_ was further checked by CV, with the curves illustrated in Figure 4C. It can be seen that a new reduction peak appeared at −0.467 V, and the reduction peak current increased gradually with the nitrite concentration. The catalytic mechanism of Nafion/HRP/SWCNTs-BP/CILE to nitrite was expressed as follows [53,54]:NO_2_^−^ + H^+^ ↔ HNO_2_
3HNO_2_ → 2NO +NO_3_^−^ + H^+^ + H_2_O
HRP heme Fe(III) + H^+^ + e^−^ ↔ HRP heme Fe(II)
HRP heme Fe(II) + NO → HRP heme Fe(II)-NO
HRP heme Fe(II)-NO + H^+^ + e^−^ → HRP heme Fe(II) + product + H_2_O

In the concentration range from 0.8 to 49.6 mmol/L, the reduction current and NaNO_2_ concentration exhibited a good linear relationship, with the regression equation as Iss (mA) = 0.0216 C (mmol/L) + 0.2310 (γ = 0.992) (Figure 4D), and the detection limit was 0.27 mmol/L (3σ). When the concentration of NaNO_2_ was more than 49.6 mmol/L, the reduction peak value reached a stable state, demonstrating a Michaelis–Menten kinetic mechanism for the electrocatalytic reaction of NaNO_2_ [55]. Based on the Lineweaver–Burk equation [56], the apparent Michaelis–Menten constant (*K_M_^app^*) could be calculated as 8.37 mmol/L.

The electrocatalytic reduction of H_2_O_2_ was also investigated by cyclic voltammetry, and the curves are shown in Figure 4E. Along with the increase in H_2_O_2_ concentration in the range 1.0–40.0 mmol/L, the linear regression equation between the reduction currents and the H_2_O_2_ concentration was Iss (mA) = 0.0122 C (mmol/L) + 0.0498 (γ = 0.991) (Figure 4F), and the detection limit was 0.33 mmol/L (3σ). The value of *K_M_^app^* for H_2_O_2_ reduction was defined as 7.88 mmol/L.

A systematic comparison with different redox protein or enzyme modified electrodes for the electrocatalytic detection of TCA, NaNO_2_, or H_2_O_2_ are listed in Table 2, which demonstrates a relatively wider linear range or lower detection limit, indicating the positive effect of the SWCNTs-BP nanocomposite on the HRP direct electrochemistry.

### 2.5. Analytical Applications

To evaluate the practical analytical applications of Nafion/HRP/SWCNTs-BP/CILE, various kinds of samples were analyzed. A medical facial peel solution was purchased from Shanghai EKEAR Bio. Tech. Co., (Shanghai, China) and diluted 10 times using pH 4.0 PBS for TCA determination. Pickled vegetables were obtained from Guilinyang supermarket (Haikou, China), which were cut into small pieces and soaked in water to obtain testing samples by filtering out of the mixture for the determination of NaNO_2_. A 3% H_2_O_2_ disinfectant was brought from Guangdong Nanguo Pharmaceutical Ltd. Co., (Guangdong, China) and diluted 100 times. As shown in Table 3, the targets could be found in the samples, and the recoveries were calculated by the standard addition method, which were determined to be in the range of 94.2–104.3%, 97.0–103.7%, and 97.7–103.0%, with the relative standard deviation (RSD) less than 3.0%, demonstrating that this electrochemical HRP sensor based on the SWCNTs-BP modified electrode could realize the detection of the real samples.

### 2.6. Stability and Reproducibility

The stability and reproducibility of Nafion/HRP/SWCNTs-BP/CILE was studied by successive CV tests in pH 4.0 PBS for 50 cycles. The redox peak currents remained almost constant and varied by 1.87% and 3.02% from the 1st to the 50th cycles, compared with the initial current. After storage in a refrigerator at 4 °C for 10 days, the reduction peak current for Nafion/HRP/SWCNTs-BP/CILE still retained 99.32% of its initial value. After 15 and 20 days of storage, the reduction peak current for the HRP-modified electrode maintained 98.37% and 94.79% of its initial value. This demonstrated that this electrode showed significant storage stability and a long life, which could be attributed to the favorable microenvironment for HRP, and the fact that the SWCNTs-BP nanocomposite was not degraded.

## 3. Materials and Methods

### 3.1. Reagents and Chemicals

The following were used as received: 1-hexylpyridinium hexafluorophosphate (HPPF_6_, Lanzhou Yulu Fine Chem. Co., Lanzhou, China), HRP (MW. 40000, Sinopharm Chem. Reagent Co., Shanghai, China); black phosphorus nanoplates (BPNPs, Nanjing XFNANO Materials Tech. Ltd. Co., Nanjing, China), purified amio single-walled carbon nanotubes (SWCNTs, diameter: 1–2 nm, length: 5–30 nm, special surface area: 380 m^2^/g, Nanjing XFNANO Materials Tech. Ltd. Co., Nanjing, China), trichloroacetic acid (TCA, Tianjin Kemiou Chem. Co., Tianjin, China), NaNO_2_ (Shanghai Chem. Plant, Shanghai, China), H_2_O_2_ (Xilong Scientific Ltd. Co., Guangzhou, China), 1-methyl-2-pyrrolidinone (NMP, 99.5%, Shanghai Aladdin Regent Ltd. Co., Shanghai, China), and Nafion (5.0%, Dupont, Wilmington, NC, USA). The supporting electrolyte was 0.1 mmol/L PBS. All other reagents were of analytical grade. Ultrapure water was obtained from a Milli-Q water purification system (Milli-Q IQ7000, Merck, Billerica, MA, USA) throughout the experiments.

### 3.2. Instruments

A CHI 1040 C electrochemical workstation (Shanghai CH Instrument, Shanghai, China) was employed for all the electrochemical experiments using a conventional three-electrode system, including the laboratory self-made Nafion/HRP/SWCNTs-BP/CILE as the working electrode, saturated calomel electrode (SCE) as the reference electrode, and a platinum wire electrode as the counter electrode. XPS was performed on an AXIS HIS 165 spectrophotometer (Kratos Analytical, Manchester, UK). Raman spectra were analyzed on a Lab RAM HR system using 532 nm lasers (Horiba, Longjumeau, France). SEM images were obtained on a JSM-7600F instruments (JEOL, Kobe, Japan), with TEM images obtained using a JEM-2010F device (JEOL, Kobe, Japan).

### 3.3. Electrochemical Measurements

Electrochemical measurements were investigated by CV in a 10.0 mL electrochemical cell containing 0.1 mol/L PBS, which was deoxygenized by highly purified nitrogen for 30 min before each experiment and maintained in a nitrogen atmosphere during the experiments. The cyclic voltammogram was recorded in the potential range from 0.2 to −0.8 V (vs. SCE) at the scan rate of 100 mV/s.

### 3.4. Preparation of SWCNTs-BP Nanocomposite

Based on the reported procedure with a little modification [67], the BP nanosheets were prepared by ultrasonic-assisted liquid-phase exfoliation and centrifugation using multilayer BPNPs with the assistance of NMP solvents. In general, 20.0 mL 1.0 mg/mL BPNPs were mixed into 5.0 mL of NMP solvents and stirred for 5 min in an oxygen-isolated environment, then subjected to continuous ultrasound for 12 h under ice cooling to obtain the BP nanosheets. After that, 0.5 mg/mL of SWCNTs was added into the BP dispersion, with the volume ratio of 1:1, and stirred for 30 min. Finally, the mixed solution was sonicated for 4 h under sealed conditions to obtain SWCNTs-BP nanocomposite dispersion, which was stored at 4 °C for the following experiment.

### 3.5. Preparation of Nafion/HRP/SWCNTs-BP/CILE

CILE was fabricated according to the methods described in previous literature [68], which was then used as a basic electrode and the electrode surface was smoothed on a weighing paper before modification. In a glove box filled with nitrogen, 10.0 μL of SWCNTs-BP suspension solution was spread on the newly prepared CILE surface and dried at room temperature to obtain SWCNTs-BP/CILE. Then, 10.0 μL of 15.0 mg/mL HRP solution was cast on the surface of SWCNTs-BP/CILE. After drying at room temperature, 10.0 μL of 0.5% Nafion solution was further decorated evenly on the surface of HRP/SWCNTs-BP/CILE to form a protective film, and then dried under nitrogen at room temperature to fabricate the modified electrode (Nafion/HRP/SWCNTs-BP/CILE). For comparison, other electrodes were prepared using a similar procedure.

## 4. Conclusions

In summary, we have developed a new electrochemical biosensor based on a SWCNTs-BP nanocomposite for investigating the electrochemical behavior of HRP. The interaction between BP with SWCNTs forms a stable composite that can prevent the degradation of BP. Due to the synergistic effects between BP and SWCNTs, the SWCNTs-BP nanocomposite exhibited high conductivity and a large specific surface area. The direct electron transfer of HRP on the SWCNTs-BP modified electrode was accelerated. Further, the biosensor displayed excellent electrochemical catalytic activity regarding the reduction of TCA, NaNO_2_, and H_2_O_2_, with a wide linear range, a low detection limit, and good electrocatalytic stability. All the results indicated that the SWCNTs-BP nanocomposite could be used for electrode modification and signal amplification. The proposed biosensor was successfully applied for the detection of real samples, with satisfactory results.

## Data Availability

Not applicable.

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
