# Peer review of "Preparation and Application of Electrochemical Horseradish Peroxidase Sensor Based on a Black Phosphorene and Single-Walled Carbon Nanotubes Nanocomposite"

_molecules, 2022, doi:10.3390/molecules27228064_

Round 1
Reviewer 1 Report
In this manuscript, the authors presented a new electrochemical sensor based on SWCNTs-BP nanocomposites for HRP immobilization with CILE as substrate electrode, and investigated the electrochemical and electrocatalytic performances. The sensor was used to detect of different targets such as TCA, NaNO2 and H2O2 with good sensitivity and high stability. The paper is well-designed with detailed electrochemical data and the improved analytical performances, which is suggested for the publication after minor revision. Followings are some problems that should be resolved.
1. To give readers a comprehensive understand of 2D nanostructure, some recent progress including Small, 2022, 18, 2104993; Anal. Chem., 2022, 94, 3669; Biosens. Bioelectron., 2021, 171, 112707 deserve citation in the part of introduction.
2. Why carbon ionic liquid electrode (CILE) is selected the bare electrode for further modification? Please elucidate the reason.
3. The abstract should be rewritten with the format as the purpose, the basic design and the major results.
4. Figure 3 is common and should be shifted to the supplementary material section.
5. Electrochemical measurement procedure should be added in Section 3.
6. English of the whole paper should be checked carefully.
Reviewer 2 Report
Dear Editor
The article “Preparation and Application of Electrochemical Horseradish Peroxidase Sensor Based on Black Phosphene and Single-Walled Carbon Nanotubes Nanocomposite” The article is relevant for scientific community but needs some improvement in some parts. I think the authors must convince the reader why using all the materials used in the modified electrode and why nitrite, H2O2 and TCA sensing.
I have comments, questions and suggestions that can enhance the performance of manuscript, enumerated below:
1. The key-words must be different from the title to get more visibility for your work.
2. In the first paragraph of Introduction, the authors have described the properties of black phosphorene in a singular way. I suggest the authors comment deeper in this topic, since the material make part of modified electrode.
3. In lines 47-48, I missed the link between the paragraphs. Improve this!
4. I missed the authors comment about the role of HRP in the electrode on the introduction section. I think the authors must perform a paragraph specially for enzyme, commenting about biosensor in general, and about the role of HRP onto an electrode surface.
5. In Fig 2D the Raman spectra of SWCNT must be together with two other Raman spectra for comparisons.
6. In Fig 3A, the authors must provide the ΔEp and the ratio of Ipa/Ipc of each modification and compared the results based on these informations.
7. What about the electroactive area of each modification? What is the enhancement of modification with BP and SWCNT compared to counterparts?
8. The authors must explain in a better way the role of HRP and the meaning of redox couple in Fig 4.
9. It is not clear for me why the authors are using HRP for nitrite and TCA sensing. H2O2 I can understand. It must be explained. What is the mechanism envolved? Do you really think the HRP is helping the electrochemical signal of nitrite and TCA?
10. Table 1 has so many works for the same authors that don’t justify. It must be made a new table with another electrode’s architectures for each analyte. It seems that authors have made the same work just changing the modifier material. Compare nitrite sensing with another works: Journal of Sol-Gel Science and Technology (2018) 87:216–229; Colloids and Surfaces A: Physicochemical and Engineering Aspects, Volume 637, 20 March 2022, 128271; Journal of Colloid and Interface Science, Volume 607, Part 2, February 2022, Pages 1313-1322
Round 2
Reviewer 2 Report
The authors have improved the manuscript, but the references need to be formatted, after that I recommend the publication in Molecules.